# Efficacy and safety of intravenous glucocorticoid therapy for IgG4-related ophthalmic disease

**Min Kyu Yang[1]◐, Gye Jung Kim[2]◐, Yeong A. Choi[1], Ho-Seok Sa◉[1]***

**1** Department of Ophthalmology, Asan Medical Center, University of Ulsan College of Medicine, Seoul, Republic of Korea, **2** Department of Ophthalmology, Myongji Hospital, Gyeonggi-do, Republic of Korea

◐ These authors contributed equally to this work.
* lineblue@hanmail.net

**Data Availability Statement:** The data contain potentially identifying information. Data are available from the Asan Medical Center Institutional Data Access / Ethics Committee (contact via

## Abstract

### Purpose

To evaluate and compare the efficacy and safety of intravenous (IV) glucocorticoid therapy with those of oral glucocorticoids as a first-line treatment for IgG4-related ophthalmic disease (IgG4-ROD).

### Methods

We retrospectively reviewed the medical records of patients who underwent systemic glucocorticoid therapy for biopsy-proven IgG4-ROD from June 2012 to June 2022. Glucocorticoids were given either oral prednisolone at an initial dose of 0.6 mg/kg/day for four weeks with subsequent tapering or once weekly IV methylprednisolone (500 mg for six weeks, then 250 mg for six weeks), according to the date of treatment. Clinicoserological features, initial response, relapse during follow-ups, cumulative doses of glucocorticoids, and adverse effects of glucocorticoids were compared for the IV and oral steroid groups.

### Results

Sixty one eyes of 35 patients were evaluated over a median follow-up period of 32.9 months. The complete response rate was significantly higher in the IV steroid group (n = 30 eyes) than in the oral steroid group (n = 31 eyes) (66.7% vs. 38.7%, $p$ = 0.041). Kaplan–Meier analysis showed that the 2-year relapse-free survival was 71.5% (95% confidence interval: 51.6–91.4) and 21.5% (95% confidence interval: 4.5–38.5) in the IV steroid and oral steroid group, respectively ($p$ < 0.001). Although the cumulative dose of glucocorticoids was significantly higher in the IV steroid group than in the oral steroid group (7.8 g vs. 4.9 g, $p$ = 0.012), systemic and ophthalmic adverse effects were not significantly different between the two groups throughout follow-ups (all $p$ > 0.05).

irb@amc.seoul.kr) for researchers who meet the criteria for access to confidential data.

**Funding:** The authors received no specific funding for this work.

**Competing interests:** The authors have declared that no competing interests exist.

## Conclusions

As a first-line treatment for IgG4-ROD, IV glucocorticoid therapy was well-tolerated, led to better clinical remission and more effectively prevented inflammatory relapse than oral steroids. Further research is needed to establish guidelines on dosage regimens.

## Introduction

IgG4-related disease (IgG4-RD) is an immune-mediated fibroinflammatory condition characterized by a dense lymphoplasmacytic infiltrate rich in IgG4-positive plasma cells and frequently elevated serum IgG4 [1,2]. IgG4-RD has been described in several organs, such as the pancreas, lymph nodes, and salivary glands [2,3]. Ocular adnexal or orbital manifestation of IgG4-RD is referred to as IgG4-related ophthalmic disease (IgG4-ROD). It is not uncommon, comprising 23% of IgG4-RD [4] and 37% of idiopathic orbital inflammation [5]. The lacrimal gland is the most frequently affected structure in patients with IgG4-ROD [6]. However, other orbital structures, such as nerves, extraocular muscles (EOM), fat, and eyelid, can also be affected [7,8].

The mainstay of the initial treatment for IgG4-ROD is the oral administration of glucocorticoids, which provide favorable responses in most cases. Nevertheless, persistent lesions or relapse of IgG4-ROD commonly occurs after initial oral steroid therapy [9,10]. Previous studies reported relapse rates of 61–79% after systemic steroids, and the re-administration or long-term maintenance of oral steroids is frequently recommended [4,11]. However, prolonged or repeated administration of oral steroids can increase the risk of glucocorticoid-induced adverse effects, non-compliance, or the need for additional immunosuppressive treatments [1]. In addition, the persistent inflammatory burden of IgG4-ROD might lead to ocular adnexal lymphoma [12]. Several factors related to treatment response or disease relapse have been investigated in previous studies [1,7,10,13]; however, an effective route for glucocorticoid administration has not been evaluated in patients with IgG4-ROD.

Thyroid-associated ophthalmopathy (TAO), another major etiology of chronic immune-mediated orbital inflammation, has been well-investigated concerning the route of glucocorticoid administration. Intravenous (IV) glucocorticoid therapy is the first-line treatment for moderate-to-severe and active TAO because several randomized trials revealed that IV steroid therapy is more effective and better tolerated than oral corticosteroids [14–16]. The present study aimed to evaluate and compare the efficacy and safety of IV glucocorticoid therapy with those of oral glucocorticoids as a first-line treatment for IgG4-ROD.

## Methods

We retrospectively reviewed the medical records of patients who underwent systemic glucocorticoid therapy as an initial treatment for IgG4-ROD. All patients were treated by one ophthalmologist (H-SS) at our institution (a 2700-bed academic tertiary referral hospital in Seoul, Korea) from June 2012 to June 2022. Only patients with biopsy-proven IgG4-ROD, who were followed up for > 3 months after completion of the initial treatment, were included in this study. Diagnosis of IgG4-ROD was based on the clinical and histopathologic criteria established by the Japanese Study Group for IgG4-ROD [17]. Exclusion criteria included the presence of concurrent extranodal marginal zone lymphoma, pregnancy during follow-up, post-treatment follow-up < 3 months, and the use of adjunctive immunosuppressants other than

glucocorticoids for initial treatment. The study was conducted according to the tenets of the Declaration of Helsinki. The protocol was approved by our Institutional Review Board (approval No. 2021–1323).

Reviewed data included detailed medical history, physical measurements, ophthalmologic examinations, laboratory tests including serum IgG4 level, and imaging studies including head and neck, chest, abdominal, and pelvic computed tomography scans, and/or positron emission tomography.

Treatments were started using systemic glucocorticoids alone in all patients. Glucocorticoids were either orally or intravenously administered, according to the date of treatment. All patients treated between June 2012 and December 2016 received oral prednisolone (Pd) alone [the oral steroid group]: oral Pd was given at an initial dose of 0.6 mg/kg/day for four weeks and then was tapered by 5 mg every 2–4 weeks. All patients treated between January 2017 and June 2022 received IV methylprednisolone (mPd) alone [the IV steroid group]: IV mPd was administered once a week for 12 weeks (500 mg weekly for six weeks, then 250 mg weekly for six weeks), with potential subsequent response-based oral Pd dose titration. In both treatment groups, low-dose oral Pd (5–10 mg/day) was maintained if persistent inflammatory signs or abnormally high levels of serum IgG4 were detected.

Patients were followed up every 1–2 months for a minimum of six months, and the interval was adjusted to 3–6 months according to the patient's condition. Serum IgG4 levels were measured every 3–6 months, before and after the initial treatment or relapse. Follow-up imaging studies were performed three months after the initial treatment and when a relapse was suspected.

Complete response was defined as complete clinical remission, with no symptoms or signs and no lesions on follow-up radiological images after initial treatment [18]. Relapse was defined as recurred inflammatory symptoms or signs with compatible findings on radiological images during follow-up. In case of relapse, patients were administered additional oral steroids, intralesional triamcinolone injection, or other immunosuppressants, such as azathioprine or methotrexate.

Systemic and ophthalmic adverse effects were assessed three months after the termination of systemic glucocorticoid therapy. Systemic adverse effects included weight gain (10% or more increase compared to the baseline weight), hypertension (systolic blood pressure $\geq 140$ mmHg) [19], diabetes (random plasma glucose $\geq 200$ mg/dL) [20], elevated liver enzyme (alkaline transaminase $\geq 100$ IU/L) [21], epigastric soreness, and insomnia. Ophthalmic adverse effects included increased intraocular pressure ($\geq 21$ mmHg) and substantial cataract progression.

We compared the response to initial treatment, change of serum IgG4 level, relapse during follow-up, and adverse effects between the oral steroid and IV steroid groups. The cumulative doses of Pd or an equivalent, calculated using the Pd conversion factor for mPd (1.25) [22], were also compared between the two groups according to the follow-up periods. Statistical analyses were performed using IBM SPSS Statistics software (version 21.0, IBM Corp., Armonk, NY, USA). The Mann–Whitney U test was used to compare continuous variables, and Fisher's exact test was used to compare categorical variables. The Kaplan–Meier method was applied for the relapse-free survival analysis of IgG4-ROD. Similar analyses were also performed in the subgroup of patients with elevated initial IgG4 levels. Multivariate Cox regression analysis, with robust standard errors that accounted for inter-eye correlation, was used to evaluate the independent association of risk factors with relapse of IgG4-ROD. A $p$ value cutoff of 0.20 was used to select covariates for inclusion in the multivariate regression. A two-sided $p$ value $< 0.05$ was considered to be statistically significant.

## Results

Sixty one eyes of 35 patients were included in the present study, with a median age of 54.1 years (interquartile range [IQR], 47.4–59.7). Among them, 30 eyes of 17 patients underwent IV mPd therapy as an initial treatment for IgG4-ROD. Baseline characteristics, such as age, sex, bilaterality, ophthalmic or systemic manifestations, and serologic features, were not significantly different between the IV and oral steroid groups (Table 1).

The median follow-up period of all patients was 32.9 months (range, 6.0–112.7). Outcomes of IV or oral glucocorticoid therapy for IgG4-ROD are presented in Table 2. The median cumulative dose of Pd or an equivalent during initial steroid treatment was significantly higher in the IV steroid group than in the oral steroid group (7.3 g vs. 1.9 g, $p < 0.001$, Mann–Whitney U test), and the rate of complete response to initial steroid treatment was also significantly higher in the IV steroid group than in the oral steroid group (66.7% vs. 38.7%, $p = 0.041$, Fisher's exact test). The rates of relapse at 1 and 2 years were significantly lower in the IV steroid group than those in the oral steroid group (9.1% vs. 41.7%, $p = 0.018$; and 30.0% vs. 95.0%, $p < 0.001$, respectively).

The median cumulative dose of glucocorticoid throughout follow-ups was significantly higher in the IV steroid group than in the oral steroid group (7.8 g vs. 4.9 g, $p = 0.012$, Mann–Whitney U test). Excluding patients who were followed for less than one year, the median cumulative doses of glucocorticoid in both groups were comparable (7.9 g vs. 7.1 g, $p = 0.187$). During follow-ups, the median cumulative dose in the oral steroid group gradually increased and became similar to the dose in the IV steroid group (Fig 1). The rate of receiving intralesional triamcinolone injection was significantly lower in the IV steroid group than in the oral steroid group (13.3% vs. 38.7%, $p = 0.016$, Fisher's exact test). No life-threatening conditions or ophthalmic adverse effects were observed. Weight gain was the most common adverse effect in both groups. No significant difference was detected between the two groups in adverse effects throughout follow-ups (all $p > 0.05$) (Table 2).

**Table 1. Baseline characteristics of patients who received intravenous (IV) or oral glucocorticoid therapy for IgG4-related ophthalmic disease.**

| n = 61 eyes (35 patients) | IV steroid group (n = 30 eyes, 17 patients) | Oral steroid group (n = 31 eyes, 18 patients) | p value |
|---|---|---|---|
| Age, years | 53.0 (48.6–59.6) | 55.3 (47.1–60.0) | 0.613 |
| Male sex, patients | 10 (58.8) | 9 (50.0) | 0.738 |
| Symptom duration, months | 6 (3–12) | 6 (2–13) | > 0.99 |
| Bilateral involvement, patients | 13 (76.5) | 13 (72.2) | > 0.99 |
| Ophthalmic manifestations | | | |
| Lacrimal gland | 24 (80.0) | 31 (100.0) | 0.011 |
| Orbital nerve | 9 (30.0) | 3 (9.7) | 0.059 |
| Extraocular muscle | 3 (10.0) | 1 (3.2) | 0.354 |
| Systemic manifestations, patients | 13 (76.5) | 12 (66.7) | 0.711 |
| Head and neck | 13 (76.5) | 12 (66.7) | 0.711 |
| Others | 6 (35.3) | 7 (38.9) | 0.826 |
| Serologic features, patients | | | |
| Eosinophil > 7% | 4 (23.5) | 3 (16.7) | 0.691 |
| IgG4 before treatment, g/L | 3.82 (0.41–7.51) | 0.36 (0.25–3.32) | 0.042 |
| IgG4 before treatment > 1.35 g/L | 11 (64.7) | 6 (33.3) | 0.094 |
| IgG4 after initial treatment, g/L | 0.71 (0.48–2.20) | 0.47 (0.24–1.73) | 0.178 |

Data are presented as n (%) for categorical variables and median (interquartile range) for continuous variables.

If not specified, comparisons were on a per-eye basis.

**Table 2. Outcomes of intravenous (IV) or oral glucocorticoid therapy as an initial treatment of IgG4-related ophthalmic disease.**

| n = 61 eyes (35 patients) | IV steroid group (n = 30 eyes, 17 patients) | Oral steroid group (n = 31 eyes, 18 patients) | p value |
|---|---|---|---|
| Follow-up period, months | 30.6 (10.6–34.0) | 37.9 (16.7–51.2) | 0.225 |
| Initial steroid treatment | | | |
| Duration, month | 3.9 (3.0–8.6) | 3.7 (2.0–6.0) | 0.302 |
| Cumulative dose of steroids, g | 7.3 (6.4–7.8) | 1.9 (1.4–2.3) | < 0.001 |
| Complete response to initial treatment | 20 (66.7) | 12 (38.7) | 0.041 |
| Relapse after initial steroid treatment | | | |
| At 1 year after initial treatment | 2/22 (9.1) | 10/24 (41.7) | 0.018 |
| At 2 years after initial treatment | 6/20 (30.0) | 19/20 (95.0) | < 0.001 |
| Throughout follow-ups | | | |
| Cumulative dose of steroid, g | 7.8 (7.1–9.8) | 4.9 (2.9–8.2) | 0.012 |
| Use of adjunctive immunosuppressants | 7 (23.3) | 9 (29.0) | 0.772 |
| Use of intralesional TA injection | 4 (13.3) | 12 (38.7) | 0.040 |
| Systemic adverse effects, patients | | | |
| Weight gain (10% or more increase) | 3 (17.6) | 4 (22.2) | > 0.99 |
| Cushingoid feature | 1 (5.9) | 2 (11.1) | > 0.99 |
| Diabetes (random plasma glucose > 200 mg/dL) | 1 (5.9) | 1 (5.6) | > 0.99 |
| Hypertension (SBP ≥ 140 mmHg) | 2 (11.8) | 2 (11.1) | > 0.99 |
| Elevated alkaline transaminase (≥ 100 IU/L) | 0 (0.0) | 1 (5.6) | > 0.99 |
| Insomnia | 4 (13.3) | 1 (5.6) | 0.177 |
| Ophthalmic adverse effects, patients | 0 (0.0) | 0 (0.0) | > 0.99 |

IQR = interquartile range; SBP = systolic blood pressure; TA = triamcinolone.

Data are presented as n (%) for categorical variables and median (interquartile range) for continuous variables.

If not specified, comparisons were on a per-eye basis.

Kaplan–Meier survival analysis showed that IV glucocorticoid therapy yielded a greater relapse-free survival than oral glucocorticoid therapy: The 2-year relapse-free survival was 71.5% (95% confidence interval [CI] 51.6–91.4) in the IV steroid group and 21.5% (95% CI 4.5–38.5) in the oral steroid group ($p < 0.001$) (Fig 2A). The multivariate Cox regression

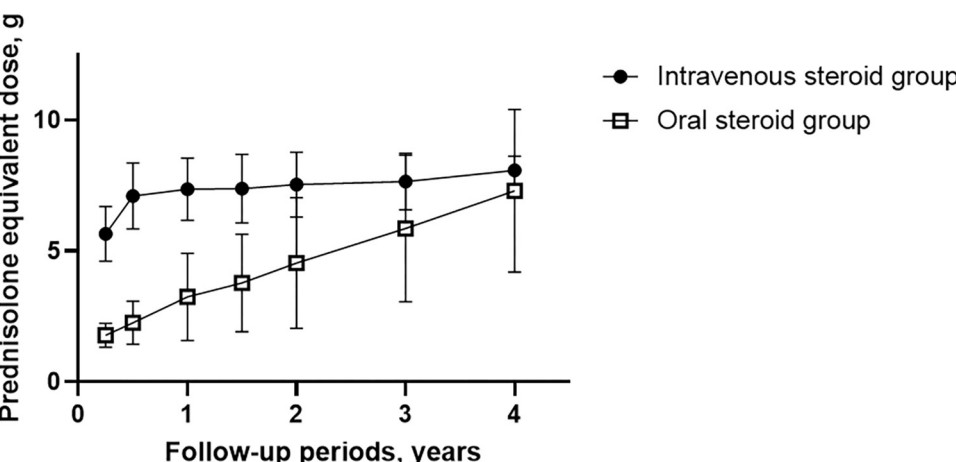

**Fig 1. The median cumulative prednisolone equivalent dose in the intravenous and oral steroid group throughout follow-ups.**

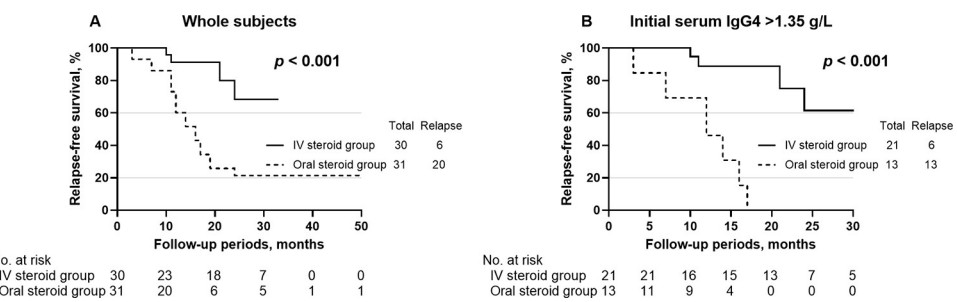

**Fig 2. Kaplan–Meier analysis of relapse-free survival for IgG4-related ophthalmic disease.** Intravenous (IV) glucocorticoid therapy yielded a greater relapse-free survival than oral glucocorticoid therapy. (**A**) In all patients (hazard ratio 5.5; 95% confidence interval 2.4–12.8; $p < 0.001$). (**B**) In the subgroups with initial serum IgG4 level $> 1.35$ g/L (hazard ratio 19.2; 95% confidence interval 6.1–60.9; $p < 0.001$).

analysis revealed that IV administration of glucocorticoids was independently associated with a significant reduction in the relapse of IgG4-ROD (adjusted hazard ratio 0.236; 95% CI 0.081–0.690; $p = 0.008$) compared with that observed with the oral administration of glucocorticoids (Table 3).

In the subgroups with elevated initial serum IgG4 levels, 21 eyes (11 patients) and 13 eyes (7 patients) were included in the IV and oral steroid subgroups, respectively. No statistically significant difference was detected in the serum levels between the IV and oral subgroups (7.0 g/L vs. 5.1 g/L, $p = 0.084$). However, the rate of complete response to initial steroids was significantly higher in the IV steroid subgroup than in the oral steroid subgroup (66.7% vs. 15.4%, $p = 0.005$). Kaplan–Meier survival analysis showed that IV glucocorticoid therapy yielded a greater relapse-free survival than oral glucocorticoid therapy in the subgroups with elevated initial serum IgG4 levels (hazard ratio 19.2; 95% CI 6.1–60.9; $p < 0.001$) (Fig 2B).

## Discussion

This comparative study found that IV glucocorticoid therapy, as a first-line treatment, resulted in more rapid and sustained clinical remission of IgG4-ROD than oral therapy. Treatment

**Table 3. Univariate and multivariate hazard ratios and 95% confidence intervals of risk factors for relapsed IgG4-related ophthalmic disease.**

| Variables | Hazard ratio (95% CI) | *p* value |
|---|---|---|
| Univariate analysis | | |
| Male sex | 0.382 (0.131–1.116) | 0.078 |
| Symptom duration | 0.971 (0.904–1.041) | 0.395 |
| Bilateral involvement | 1.557 (0.518–4.681) | 0.431 |
| Systemic involvement | 0.662 (0.234–1.870) | 0.436 |
| Orbital nerve or EOM involvement | 0.825 (0.244–2.792) | 0.757 |
| Eosinophilia (>7% of leukocytes) | 0.789 (0.204–3.055) | 0.731 |
| Elevated serum IgG4 before treatment[a] | 2.222 (0.696–7.094) | 0.177 |
| IV glucocorticoid therapy (vs. oral therapy) | 0.236 (0.081–0.690) | 0.008 |
| Complete response to initial treatment | 0.382 (0.120–1.210) | 0.102 |
| Multivariate analysis | | |
| IV glucocorticoid therapy (vs. oral therapy) | 0.236 (0.081–0.690) | 0.008 |

CI = confidence interval; EOM = extraocular muscle; IV, intravenous.

[a]Serum IgG4 level $> 1.35$ g/L.

response to initial steroid treatment was better in the IV group than in the oral group. The IV steroid group also showed better relapse-free survival during subsequent follow-ups. Although the cumulative steroid dose was higher in the IV steroid group, IV steroids were safe, with adverse effects at the same frequency as observed with oral steroids.

The response rates in our study (67% and 39% in the IV and oral groups, respectively) may appear lower than those reported in previous studies (66–100%) [1,10,23]. This may be due to our strict definition of complete response, including complete clinical and radiological remission after only initial treatment for four months. The better response in the IV group than in the oral group can be associated with a higher initial dose of glucocorticoids. Although the precise mechanism is still unclear, the abnormal adaptive immune responses mediated by T helper type 2 cells, regulatory T lymphocytes, and cytotoxic T lymphocytes may be involved in the IgG4-RD pathogenesis [24]. In vitro studies showed that glucocorticoids induce T helper type 2 cell apoptosis and suppress T cell-mediated cytotoxicity in dose-dependent manners [25,26]. Randomized clinical trials for the treatment of autoimmune disorders, such as giant cell arteritis and TAO, have also demonstrated that high-dose IV steroid therapy yields a better clinical response than oral steroid therapy [27,28]. Oral administration of high-dose glucocorticoids equivalent to IV therapy may not be tolerated due to adverse events, such as gastrointestinal problems and weight gain [27,29]. To the best of our knowledge, IV steroid therapy for IgG4-ROD has been limitedly applied to severe cases with optic nerve dysfunction [30–32]. Our results and previous reports suggest that further structured studies are required to evaluate the efficacy of IV steroid therapy for a wider disease spectrum of IgG4-ROD.

In addition to improving the treatment response, IV steroid therapy appears to be beneficial in preventing long-term relapse. This may be associated with a better initial response due to a high dose of glucocorticoids [33] or the effect of the pulse regimen. A previous study reported that IV high-dose steroid therapy completely abolished circulating dendritic cells that play a central role in initiating the primary immune response [34]. A higher initial dose of glucocorticoid may induce epigenetic changes, leading to long-term suppression of relapse [35]. Sugimoto et al. reported that the 5-year relapse-free rates of autoimmune pancreatitis were higher in the IV steroid group than in the oral steroid group with marginal significance (77.8% vs. 46.9%, $p = 0.09$), while the cumulative doses of glucocorticoids were comparable (3.7 g vs. 4.2 g, $p = 0.33$) [36]. Additional studies with various doses and schedules will help clarify the long-term effects of the steroid regimen.

Considering the chronic progressive nature of IgG4-ROD, reducing the risk of relapse has several clinical advantages. Previous studies have shown that IgG4-ROD relapse is a common problem after oral steroid therapy [4,11], leading to a need for repeated treatments or long-term maintenance of oral steroids [13]. Repeated oral steroid therapy can induce non-compliance and psychological stress in patients compared to IV steroid therapy following a predetermined schedule. Chronic administration of oral steroids is known to deteriorate quality of life in patients with recurrent TAO [37,38]. The emergence of mutated plasmablast clones in patients with relapsed IgG4-RD [39] implies that the recurrent IgG4-ROD might be associated with an increased risk of developing ocular adnexal lymphoma, a possible complication of long-standing IgG4-ROD [12].

Serological factors, including high serum IgG4 level, elevated serum IgE, and elevated eosinophils, are known to be associated with IgG4-ROD relapse [7,40]. In our study, serum IgG4 levels before initial steroid therapy were significantly higher in the IV group than in the oral group. However, our IV steroid regimen resulted in a lower relapse rate than that with oral steroids. In the subgroup analysis of only patients with elevated serum IgG4 levels, the IV group still showed a significantly better complete response rate (67% vs. 15%) and relapse-free survival than the oral group, as shown in Fig 2B.

A high steroid dose in the IV steroid group may raise concerns about the adverse effects [22]. However, as the cumulative dose of glucocorticoids in the oral steroid group gradually increased due to more frequent relapses, the difference in cumulative dose of glucocorticoids between the two groups decreased during follow-ups. In addition, the further requirement of steroids may be underestimated in the oral steroid group when patients had early follow-up loss, which can be supported by comparable cumulative steroid doses of patients who were followed for one year or more in both groups. As a result, the rates of adverse effects in the IV steroid group were not different from those in the oral steroid group. Several studies on steroid therapy for TAO described weight gain and cushingoid features in a high proportion of patients on oral steroids [27,29]. A similar observation was documented in our study, although not statistically significant compared with the IV group. Regarding the dose and frequency of IV steroids for IgG4-ROD, we employed the most common regimen of IV mPd used for active TAO [14], and the cumulative dose of glucocorticoids did not exceed the recommended dose range (Pd equivalent < 10 g) in the present study. There were no cases of hepatitis or liver failure throughout follow-ups in our cohort, which can be associated with IV high-dose steroids [21]. These findings suggest that the initial administration of high-dose steroids is well-tolerated and safe for most patients.

Our study has several limitations. First, in the retrospective study, the route of steroid administration was determined according to the date of treatment, and we could not fully randomize the characteristics of patients, such as serum IgG4 level. However, given the negative prognostic effect of increased serum IgG4 level, it does not seem to undermine the usefulness of IV steroids revealed in this study. Second, the number of patients and follow-up periods were insufficient to draw firm conclusions. Also, we could not perform a stratified analysis according to disease severity due to the lack of a quantifying system of IgG4-ROD severity. Third, we employed the IV steroid protocol used in patients with TAO (mPd, 500 mg, then 250 mg, six weeks each) for the treatment of IgG4-ROD. Further long-term, randomized clinical trials are required to determine the optimal dose and frequency of IV glucocorticoids in patients with IgG4-ROD.

In conclusion, IV glucocorticoid therapy is an effective and safe option as a first-line treatment for IgG4-ROD; it can lead to better clinical remission and more effectively prevent inflammatory relapse than oral steroids. A further prospective study is needed to clarify long-term benefits and establish a proper guideline on the dosage regimen.

## Author Contributions

**Conceptualization:** Ho-Seok Sa.

**Data curation:** Min Kyu Yang, Gye Jung Kim, Ho-Seok Sa.

**Formal analysis:** Min Kyu Yang, Yeong A. Choi.

**Supervision:** Ho-Seok Sa.

**Validation:** Ho-Seok Sa.

**Writing – original draft:** Min Kyu Yang, Gye Jung Kim.

**Writing – review & editing:** Ho-Seok Sa.

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
