## [Decision Letter · Decision Letter 0]

30 Jan 2023

PONE-D-23-01006Efficacy and safety of intravenous glucocorticoid therapy for IgG4-related ophthalmic diseasePLOS ONE

Dear Dr. Sa,

Thank you for submitting your manuscript to PLOS ONE. After careful consideration, we feel that it has merit but does not fully meet PLOS ONE’s publication criteria as it currently stands. Therefore, we invite you to submit a revised version of the manuscript that addresses the points raised during the review process.

We look forward to receiving your revised manuscript.

Kind regards,

Keiko Hosohata, Ph.D.

Academic Editor

PLOS ONE

Journal Requirements:

Additional Editor Comments:

This manuscript is interesting; however, there are few points to be corrected.

Importantly, comparisons per patient is confusing; systemic manifestations are compared on a per patient basis, but ‘Head and neck’ was not?

Reviewers' comments:

Reviewer's Responses to Questions

**Comments to the Author**

1. Is the manuscript technically sound, and do the data support the conclusions?

Reviewer #1: Yes

Reviewer #2: Yes

2. Has the statistical analysis been performed appropriately and rigorously? 

Reviewer #1: No

Reviewer #2: Yes

3. Have the authors made all data underlying the findings in their manuscript fully available?

Reviewer #1: No

Reviewer #2: Yes

4. Is the manuscript presented in an intelligible fashion and written in standard English?

Reviewer #1: Yes

Reviewer #2: Yes

5. Review Comments to the Author

Reviewer #1: Auhtors should address the following issues:

1) Abstract: Give sample size (n) for the oral steroid group in the “The complete response rate was significantly higher in the IV steroid group (n = 30 eyes) than in the oral steroid group (66.7% vs. 38.7%, p = 0.041)” sentence.

2) Table-1:

a. Although IQR is explained in table footnotes, there is no ‘IQR’ abbreviation on the table.

b. Comparisons per patient is confusing. For example systemic manifestations are compared on a per patient basis, but ‘Head and neck’ was not?

3) Table-2: Is the follow-up period comparison based on per patient or per eye?

4) Are the Cox models for relapse based on per eye or per patient?

5) If the Cox models are based on per eye, how was the lack of independency between the two eyes from the same patient were accounted for in the models?

6) Small KM and Cox analysis table presenting total patients/eyes (whichever used) at risk, total number of events/relapses observed for both groups, and hazard ratio estimates from the Cox models on Figure-2 A and B will be very useful.

Reviewer #2: The authors applied IV glucocorticoid therapy which is for moderate-to-severe TAO for IgG4-ROD.

As the cumulative dose of steroid was significantly higher in the IV steroid group than in the oral steroid group, such important message should be emphasized in the paper as well as the title.

6. PLOS authors have the option to publish the peer review history of their article (what does this mean?). If published, this will include your full peer review and any attached files.

Reviewer #1: No

Reviewer #2: **Yes: **Jian Zhu

---

## [Author Response · Author response to Decision Letter 0]

22 Mar 2023

Reviewer #1: Auhtors should address the following issues:

1) Abstract: Give sample size (n) for the oral steroid group in the “The complete response rate was significantly higher in the IV steroid group (n = 30 eyes) than in the oral steroid group (66.7% vs. 38.7%, p = 0.041)” sentence.

“(n = 31 eyes)” are added just behind the “in the oral steroid group”. 

Abstract, Line 36: 

The complete response rate was significantly higher in the IV steroid group (n = 30 eyes) than in the oral steroid group (n = 31 eyes) (66.7% vs. 38.7%, p = 0.041).

2) Table-1:

a. Although IQR is explained in table footnotes, there is no ‘IQR’ abbreviation on the table.

The table footnote explaining IQR is deleted. 

b. Comparisons per patient is confusing. For example systemic manifestations are compared on a per patient basis, but ‘Head and neck’ was not?

Systemic manifestations and serologic features are compared on a per-patient basis. Sorry for confusing mistakes on numbers. We added “If not specified, comparisons were on a per-eye basis.” in table legend. 

Methods, Line 146/162:

If not specified, comparisons are on a per-eye basis.

3) Table-2: Is the follow-up period comparison based on per patient or per eye?

Follow-up periods were calculated on a per-eye basis (Except for adverse effects comparisons, most comparisons were on a per-eye basis). 

4) Are the Cox models for relapse based on per eye or per patient?

The Cox models for relapse were on a per-eye basis. 

5) If the Cox models are based on per eye, how was the lack of independency between the two eyes from the same patient were accounted for in the models?

Thank you for the good comment. In consultation with the statistics team of our institute, additional analysis considering inter-eye correlation was performed. The results remained unchanged that IV administration of glucocorticoids was associated with a significant reduction in the relapse of IgG4-ROD.

Methods, Line 131: 

Multivariate Cox regression analysis, with robust standard errors that accounted for inter-eye correlation, was used to evaluate the independent association of risk factors with relapse of IgG4-ROD.

6) Small KM and Cox analysis table presenting total patients/eyes (whichever used) at risk, total number of events/relapses observed for both groups, and hazard ratio estimates from the Cox models on Figure-2 A and B will be very useful.

We added tables presenting number of eyes at risk/total number and relapses in both groups. Hazard ratio estimates are presented in figure legends due to the space issue. 

Results, line 195: 

(A) In all patients (hazard ratio 5.5; 95% confidence interval 2.4–12.8; p < 0.001). (B) In the subgroups with initial serum IgG4 level > 1.35 g/L (hazard ratio 19.2; 95% confidence interval 6.1–60.9; p < 0.001).

Reviewer #2: The authors applied IV glucocorticoid therapy which is for moderate-to-severe TAO for IgG4-ROD.

As the cumulative dose of steroid was significantly higher in the IV steroid group than in the oral steroid group, such important message should be emphasized in the paper as well as the title.

Thank you for the good comment. In the abstract, the treatment regimen of oral or IV methylprednisolone was additionally described in detail. 

Abstract, line 29~31: 

Glucocorticoids were orally or intravenously administered, according to the date of treatment. � Glucocorticoids were given either oral prednisolone at an initial dose of 0.6 mg/kg/day for four weeks with subsequent tapering or once weekly IV methylprednisolone (500 mg for six weeks, then 250 mg for six weeks), according to the date of treatment.

---

## [Editor Report · Decision Letter 1]

3 Apr 2023

Efficacy and safety of intravenous glucocorticoid therapy for IgG4-related ophthalmic disease

PONE-D-23-01006R1

Dear Dr. Sa,

We’re pleased to inform you that your manuscript has been judged scientifically suitable for publication and will be formally accepted for publication once it meets all outstanding technical requirements.

Kind regards,

Keiko Hosohata, Ph.D.

Academic Editor

PLOS ONE
---

## [Editor Report · Acceptance letter]

11 Apr 2023

PONE-D-23-01006R1 

Efficacy and safety of intravenous glucocorticoid therapy for IgG4-related ophthalmic disease 

Dear Dr. Sa:

I'm pleased to inform you that your manuscript has been deemed suitable for publication in PLOS ONE. Congratulations! Your manuscript is now with our production department. 

Kind regards, 

on behalf of

Dr Keiko Hosohata 

Academic Editor

PLOS ONE